# Improving the Wear and Corrosion Resistance of Aeronautical Component Material by Laser Shock Processing: A Review

**DOI:** 10.3390/ma16114124

**Published:** 2023-06-01

**Authors:** Jiajun Wu, Zhihu Zhou, Xingze Lin, Hongchao Qiao, Jibin Zhao, Wangwang Ding

**Affiliations:** 1College of Engineering, Shantou University, Shantou 515063, China; 21zhzhou@stu.edu.cn (Z.Z.);; 2State Key Laboratory of Robotics, Shenyang Institute of Automation, Chinese Academy of Sciences, Shenyang 110016, China; hcqiao@sia.cn (H.Q.);; 3Institute for Advanced Materials and Technology, University of Science & Technology Beijing, Beijing 100083, China

**Keywords:** laser shock processing, aeronautical components, wear resistance, corrosion resistance

## Abstract

Since the extreme service conditions, the serious failure problems caused by wear and corrosion are often encountered in the service process for aeronautical components. Laser shock processing (LSP) is a novel surface-strengthening technology to modify microstructures and induce beneficial compressive residual stress on the near-surface layer of metallic materials, thereby enhancing mechanical performances. In this work, the fundamental mechanism of LSP was summarized in detail. Several typical cases of applying LSP treatment to improve aeronautical components’ wear and corrosion resistance were introduced. Since the stress effect generated by laser-induced plasma shock waves will lead to the gradient distribution of compressive residual stress, microhardness, and microstruture evolution. Due to the enhancement of microhardness and the introduction of beneficial compressive residual stress by LSP treatment, the wear resistance of aeronautical component materials is evidently improved. In addition, LSP can lead to grain refinement and crystal defect formation, which can increase the hot corrosion resistance of aeronautical component materials. This work will provide significant reference value and guiding significance for researchers to further explore the fundamental mechanism of LSP and the aspects of the aeronautical components’ wear and corrosion resistance extension.

## 1. Introduction

The aircraft engine is the power system for modern advanced aircraft [1]. At the same time, as a highly complex and precise power machine, the aircraft engine is also honored as “The crown jewel of modern industry” [2,3]. As for the modern advanced aircraft engine, the aeronautical components such as blades, guide vanes, afterburners, and casings will withstand the centrifugal stresses, thermal stresses, aerodynamic loading, vibration load, wear, and hot corrosion. These extreme service conditions will be harmful to the secure flight of aircraft. With the repaid development of the advanced aircraft, the properties requirements for the aeronautical components are gradually becoming higher and higher, the service conditions will be more extreme such as the increasing working temperatures inside the aircraft engine, more serious centrifugal stresses and various loads, the aeronautical components need to fulfill the requirement of a high thrust-weight ratio for the new generation advanced aero-engines [4,5]. Wear and corrosion are the most severe failure problems often encountered in the service process for aeronautical components [6]. It can easily lead to the cycle slip, plastic accumulation and fatigue cracks for aeronautical components, resulting in the serious aircraft failure or even flight accident [7,8]. Generally, these fatigue failures are caused by wear and corrosion, originating from materials’ surface and extending to the inside, resulting in the serious failure for various components [9,10]. According to the existing literature reports, mechanical surface-strengthening technologies have been extensively investigated and applied to adjust the surface integrity parameters such as enhanced microhardness, induced compressive residual stress for the aeronautical components materials without changing any mechanical properties of basal materials, which have essential engineering application significance for reducing the rate of the growth of cracks and even the closing of cracks, and the prevalence of the serious failures of aero-engine components caused by wear and corrosion [2,4,11].

The common mechanical surface-strengthening technologies mainly comprise shot peening (SP) [12], ultrasonic rolling surface strengthening (URSS) [13], ultrasonic shot peening (USP) [14], low plasticity burnishing (LPB) [15], water jet peening (WJP) [16], ultrasonic nano structure modification (UNSM) [17] and so on. The mutual feature of mechanical surface-strengthening technologies is introducing the beneficial compressive residual stress layer, enhancing microhardness, and leading to the microstructure refinement in near-surface layer of targeted material without changing any chemical structures and properties [7]. So these mechanical surface-strengthening technologies can bring about good results and reap some useful benefits for the metallic components. However, the depth of beneficial compressive residual stress layer introduced by these common mechanical surface-strengthening technologies can only be reached to 0.3 mm at most in general, which cannot meet the ever-increasing requirement for the high properties materials and advanced modern equipment [18]. So it’s request researchers need to develop and apply more advanced mechanical surface-strengthening technology to tackle the current challenges and problems.

As one of the greatest inventions of natural science in 20th century, the laser has been applied in many engineering fields due to its valuable and exceptional performance. At the same time, many new or novel laser processing technologies utilizing laser are emerged, and these processing technologies have brought great changes to the engineering fields. Laser shock processing (LSP) is a novel mechanical surface-strengthening technology with excellent strengthening effect, controllability, and adaptability [7,8,19,20]. Compared with the common mechanical surface supporting technologies such as SP, URSS, USP and WJP, the adequate beneficial compressive residual stress layer depth induced by LSP treatment can reach over 1 mm. In addition, LSP treatment can obtain more smooth surface morphology relative; that is, LSP can obtain a lower macroscopic surface roughness. For instance, the superalloy FGH97 experimental samples were subjected to LSP treatment and SP treatment; the adequate depth of the compressive residual stress layer of FGH97 caused by SP is about 0.28 mm, while the effective depth of compressive residual stress layer of FGH97 induced by LSP is about 0.84 mm [21]. Gill et al. [22], investigated LSP, CSP, and UNSM on the residual stress of Ni-based superalloy IN718 SPF. And the related experimental results showed that the compressive residual stress layer depth induced by LSP treatment could reach about 0.6 mm, while that caused by the CSP and UNSM treatment are below 0.3 mm. Apart from residual stress, surface topography is also an important parameter to evaluate the qualities of the metallic components and materials treated by mechanical surface-strengthening technologies. Aluminum alloy A356 was modified by LSP treatment and SP treatment. The surface roughness for LSP experimental sample is about 1.1 μm, less than that of SP experimental sample with a value of 5.8 μm [22,23]. So applying LSP treatment to tackle the challenge of the serious failures of aeronautical components can be regarded as a special significance.

LSP treatment has showed evident technical advantages in the fields of surface-strengthening of aeronautical components and related materials. At present, the associated researches mainly focus on the improvements about the fatigue and residual stress of aeronautical components [24,25,26,27]. However, the researches, especially the associated summaries about wear and corrosion resistance, are rare in relative. So it is necessary to summary the related researches about the wear and corrosion resistance improvements of aeronautical components by LSP. This work summarized the fundamental mechanism of LSP in detail, and introduced several typical cases about apply LSP treatment to improve the wear and corrosion resistance of aeronautical components materials. This work will provide fundamental reference values and guiding significance for researchers to further explore the primary mechanism of LSP and the aspects of the aeronautical components’ wear and corrosion resistance extension.

## 2. Fundamental Mechanism of Laser Shock Processing

As the latest peening technology was initially introduced in the engineering fields, LSP aims to improve the mechanical performances of aeronautical components [28]. Figure 1 shows the common fundamental mechanism schematic of LSP [29]. Based on the physical interaction process between the pulsed laser and metallic material, LSP utilizes the stress effect generated by high-pressure laser-induced plasma shock waves to treat the materials’ surface [30]. When a nanosecond pulsed laser beam irradiates the surface of metallic material with the power density of GW/cm^2^ level, the surface layer of the metallic target will absorb the pulsed laser energy and cause explosive vaporization [23]. The vaporized particles will continue to absorb the pulsed laser energy and induce the plasma plume formation at high temperatures (over 10,000 K) nearly simultaneously [30]. Since the confined ablation mode in the LSP process, the laser-induced plasma will expand rapidly and form super-high-pressure shock waves (GPa level) acting on the metal target’s surface and propagating to the inside material [2,30,31,32].

As shown in Figure 1, the LSP treatment process adopts the confined ablation mode with a transparent constrained layer and an absorbing protective layer in general. The shocked surface must be coated with absorbing protective layer such as black paint, black tape, or aluminum foil, and then covered with a thin transparent constrain layer such as water or optical glass [30]. It is well known that, the temperature for lasers in engineering fields is very high. So the interaction between laser and material will inevitably lead to the laser thermal effect, which harms the properties improvement of metallic materials. So applying the absorbing protective layer can protect the metal target from laser thermal ablation and create a pure mechanical stress effect in the metallic material [2,17]. The plasma plume generated by the interaction of the laser and material contains enormous energy, which will further induce the formation of shock waves. To improve the pressure of laser-induced plasma shock waves, Clauer et al. [33], proposed the confined ablation mode, the black coating layer coated on the shocked surface of the metal target and then covered with a transparent constrain layer. The transparent constrain layer can be considered a confined space, preventing the plasma from expanding extensively [23]. As a result, the pressure of laser-induced plasma shock wave can be increased by up to two orders of magnitude compared to the plasma generated in the vacuum (without constraint layer) [2,34]. Apart from increase of laser-induced plasma shock wave pressure, the action time of the shock wave can also be increased due to the restrictive effect of the constraint layer [35]. According to Fabbro’s theory [32], the action time for the laser-induced plasma shock wave is about 2–3 times of the pulsed width in general. Since the pulsed width for the laser generally exceeds 30 ns, the laser-induced plasma shock waves’ action time is below 100 ns. Hence, the strain rates for the LSP process can be reached to the magnitude of 10^7^ s^−1^. And the interaction mechanism of laser-induced plasma shock wave and target material can be referred to the shock wave dynamics. Only when the pressure of the induced shock wave exceeds the dynamic yield strength (*σ_Y_^dyn^*) in a one-dimensional stress state, the severe plastic deformation (SPD) of the metallic target can occur. When the internal stress over the Hugoniot elastic limit (*HEL*), the one-dimensional strain state will be occurred [36]. According to the theory of Johnson and Rhode, the dynamic yield strength and *HEL* can meet the following equations [37].
(1)$$HEL=\sigma_{Y}^{dyn}\frac{1-v}{1-2v}$$
where *ν* is the Poisson’s ration. The dynamic yield strength can be computed by the Cowper-Symond constitutive equation.
(2)$$\sigma_{Y}^{dyn}=\sigma_{Y}^{sta}\left[1+{\left(\frac{\varepsilon ’}{D}\right)}^{\frac{1}{q}}\right]$$
where *σ_Y_^sta^* is the static yield strength, *D* and *q* are the constants of material, and *ε’* is strain rates, which can be expressed as follows.
(3)$$\varepsilon ’=\frac{1}{t_{0}}$$
where *t*_0_ is the action time of the stress wave.

During the LSP process, the deformation induced by LSP generated stress effect can be considered to exist in a one-dimensional stress state, and the surface of the metallic target exists in a one-dimensional strain state [36]. Since the peak pressure of the laser-induced plasma shock waves can be reached to GPa magnitude, which is higher than the dynamic yield strength, SPD will arise in the near-surface layer of metal targets and accompany by the stress effect [38]. The stress effect induced by LSP treatment can lead to cold working in the microstructure refinement evolution and the introduction of beneficial compressive residual stress [2]. Compared with other laser processing technologies, LSP treatment is characterized by using the laser-generated stress effect rather than laser thermal effect; that is LSP utilizes the stress effect of laser-induced plasma shock waves to form a gradient distribution of compressive residual stress, microhardness, and microstructure in the near-surface layer of the metallic target, thereby improving the fatigue performance, wear resistance, corrosion resistance, and tensile property, etc. [7,8].

The LSP process parameters mainly consists of laser parameters (pulsed laser energy, wavelength, laser pulse duration, laser beam spot diameter, and overlap rate), absorbing protective layer, and constrain layer. Generally, the absorbing protective layer and constrain layer is determined in most application. So the researchers usually adjust the laser parameters to obtain the desired LSP effects or results. The laser power density can be expressed as follows.
(4)$$I=\frac{4E}{\pi d^{2}\tau }$$
where *I* is laser power density, *E* is pulsed laser energy, *τ* is laser pulse duration. So it can be indicated that laser power density is the comprehensive parameter for laser parameters.

According to laser power density, the peak pressure of laser-induced plasma shock wave can be expressed as follows.
(5)$$P=0.01\sqrt{\frac{\xi }{3+2\xi }}\sqrt{Z}\sqrt{I}$$
where *ξ* is fraction of absorbed energy in the range between 0.1 and 0.5, *Z* is the reduced acoustic impedance, which can be expressed as follows.
(6)$$\frac{2}{Z}=\frac{1}{Z_{1}}+\frac{1}{Z_{2}}$$
where *Z*_1_ and *Z*_2_ are acoustic impedance for targeted material and constrained layer, respectively. The acoustic impedance can be calculated by the following equation.
(7)$$Z_{i}=\rho_{i}D_{i}$$
where *i =* 1 or 2, *ρ_i_* and *D_i_* denotes mass density and velocity of laser-induced plasma shock wave, respectively.

According to the above equations, the peak pressure of laser-induced plasma shock wave is proportional to laser power density. With the higher laser power density, which will lead to higher peak pressure of laser-induced plasma shock waves and more severe plastic deformation. As a result, the compressive residual stress, microhrdness, and the degree of grain refinement will be increased, which means that the better LSP effect. However, the excessive laser power density always lead to laser ablation phenomenon for absorbing protective layer, which will reduce the LSP effect and lead to the surface thermal ablation. So it should be select suitable laser parameters to obtain the optimal LSP effect.

Apart from laser power density, the overlap is an import parameter to affect the LSP effect. The schematic of overlap rate is shown in Figure 2. And the overlap rate can be expressed as follows:(8)$$\eta =\frac{\delta }{2R}\times 100\%=\frac{\delta }{d}\times 100\%$$
where *η* is the overlap rate (0% ≤ *η* < 100%), *δ* is the overlap distance between two adjacent laser spots, *R* is the radius of laser beam spot, *d* = 2*R*.

Generally, the higher overlap rate can be regarded as the more shocked times in the same LSP region. So the LSP effect such as induced compressive residual stress, microardness will be increased with overlap rate. Apart from the enhanced mechanical properties, the distribution of surface mechanical properties will be become more uniform. For instance, the effect of overlap rate on the residual stress distribution was investigated by Hu et al. [39], and the related results is shown in Figure 3. The experimental results showed that the increasing the overlap rate can improve the residual stress and also make it more uniform. So select a higher overlap rate can be regarded as a good means to obtain a better LSP effect.

## 3. LSP Treatment for Wear and Corrosion Resistance Improvement

The serious failures of aeronautical components due to wear and corrosion are most common phenomenon in aerospace industry. The common feature for these failures are originating from the surface and subsurface of materials. As an advanced mechanical surface-strengthening technology, LSP treatment shows tremendous advantages in improving mechanical performances for metallic materials and components. So applying LSP treatment to enhance wear and corrosion resistance can significantly solve failure problems for the aeronautical details.

### 3.1. Wear Resistance Improvement for Aeronautical Components Material

Since the operation speed of an aero-engine is very high, so the wear problems among the aeronautical components are inevitable [40]. Therefore, improving the wear resistance, especially the impact wear resistance for aeronautical components and related materials by post-treatment is very important.

Since the excellent performance, such as low mass density, high specific strength and excellent wear resistance, Ti-6Al-4V titanium alloy is widely used in the aerospace industry [41]. In the aerospace industry, the blades of aircraft engines are constantly subjected to heavy wear conditions caused by dusts, sands, and other particles entrained by airflow [42]. And there are many investigations have studied the wear mechanism and response of Ti-6Al-4V alloy by different surface-strengthening technologies. In the aspect of LSP, the impact wear behavior of Ti-6Al-4V alloy subjected to LSP treatment was investigated by Yin et al. [41]. In this LSP experiment, the laser parameters used were 5 J and 7 J for pulsed laser energy, 1064 nm for wavelength, 3 mm for laser spot diameter, 10 ns for laser pulse duration, and 33% for overlap rate. According to given laser parameters and Equation (4), the laser power density for 5 J LSP treatment and 7 J LSP treatment are 7.08 GW/cm^2^ and 9.91 GW/cm^2^, respectively. After the LSP treatment, the impact wear test with initial impact velocities of 60 mm/s and 150 mm/s was conducted for these experimental samples (before and after LSP treatment). The 3D-profile micrographs of the worn scars of experimental samples in different impact wear test conditions are shown in Figure 4. The width and maximum wear depth of the worn scars for the experiment samples in other impact wear test conditions are presented in Figure 5.

At the determined initial impact velocities, the width and maximum wear depth of worn scars have similar change laws. The lower width and maximum wear depth of worn scars means a better wear resistance. With the initial impact velocity of 60 mm/s, the width of worn scars for the experiment samples before LSP treatment (untreated), and with 5 J LSP treatment, and 7 J LSP treatment are 335.7 μm, 318.7 μm, and 280.1 μm, respectively. While the maximum wear depth of worn scars for the experiment samples with prior LSP treatment, 5 J LSP treatment, and 7 J LSP treatment are 2.0 μm, 1.79 μm, and 280.1 μm, respectively. When with the initial impact velocity was increased to 150 mm/s, both the width and the maximum wear depth of worn scars were increased. The width of worn scars for the experiment samples with prior LSP treatment, 5 J LSP treatment, and 7 J LSP treatment are increased to 479.2 μm, 448.4 μm and 409.7 μm, respectively. While the maximum wear depth of worn scars for the experiment samples with prior LSP treatment, 5 J LSP treatment and 7 J LSP treatment are increased to 4.3 μm, 3.25 μm and 2.58 μm, respectively. No matter the different impact wear test conditions, it can be indicated that the wear resistance of Ti-6Al-4V titanium alloy is improved after the LSP treatment, which is mainly caused by the microhardness improvement and the introduction of beneficial compressive residual stress [6,43]. And the microhardness and residual stress distribution of experimental samples are shown in Figure 6. In this work, the residual stresses was measured by the Proto-LXRD X-ray diffractometer with sin^2^ ψ method, the microhardness was measured by a standard Vickers indenter with an indentation load of 500 g and the dwell time of 10 s.

It is well known that wear resistance of metallic materials is well intimated with the microhardness, which can be expressed as follows [44].
(9)$$V=K\frac{PL}{H_{V}}$$
where *V* is the worn volume, *K* is the wear factor, *P* is the positive load during impact wear, *L* is the impact wear distance, and *H*v is the microhardness.

According to Figure 6, the microhardness of Ti-6Al-4V titanium alloy is increased after LSP treatment. So it can be indicated that the experimental sample’s worn volume will decrease after LSP treatment. Therefore, the wear resistance will be increased since the shock wave is induced in the LSP process, which will introduce beneficial compressive residual stress. As for metallic components, the compressive residual stress can release and hinder more severe plastic deformation and delay microcracks initiation and propagation, weakening the wearing effect [45].

Of course, the wear resistance improvement level is different in different LSP parameters. Interestingly, under the same initial impact velocity, the wear scars for the experimental sample with 7 J have the shortest width and the shallowest maximum depth. The 5 J LSP treated sample, and finally, the untreated sample. So the wear resistance of the experimental sample with 7 J is better than that with 5 J. The main reason is that the microhardness and induced compressive residual stress are increased through the LSP treatment. Higher laser pulsed energy can lead to higher pressure laser-induced plasma shock waves and produce a more obvious strengthening effect. As shown in Figure 4, the maximum microhardness and maximum compressive residual stress of all experimental samples are presented at the material’s surface (e.g., surface microhardness or surface compressive residual stress). In terms of microhardness, the surface microhardness of the Ti-6Al-4V titanium alloy experimental sample prior LSP treatment is about 390 HV, when with the LSP parameter of 5 J, the microhardness of the Ti-6Al-4V titanium alloy experimental sample is increased to about 440 HV, which is about 12.8% higher than the untreated sample. When the laser pulse energy is increased to 7 J, the microhardness of the Ti-6Al-4V titanium alloy experimental sample is increased to about 470 HV, which is about 6.82% higher than the 5 J LSP treated experimental sample. Regarding residual stress, its change law is similar to the microhardness. The surface compressive residual stress for the untreated experimental sample, 5 J LSP treated experimental sample, and 7 J LSP treated experimental sample are about 30 MPa, 495 MPa, and 625 MPa, respectively. All in all, the wear resistance improvement of the Ti-6Al-4V titanium alloy after the LSP was attributed to the enhancement of surface microhardness and beneficial compressive residual stress.

### 3.2. Corrosion Resistance Improvement for Aeronautical Components Material

With the rapid development of the aerospace industry, aeronautical components are usually used in extreme conditions such as increasing working temperature, growing working pressure, super-high-speed rotation, and beast corrosion environment (containing sodium, sulfur and chloride) [2,46]. Since the high-temperature strength, great anti-oxidation, and excellent hot-corrosion resistance, the Ni-based superalloy has become a super-excellent material to manufacture aeronautical components such as blades, guide vanes, afterburners, turbine, and so on [47]. Since the complex working condition, the hot-corrosion and high-temperature oxidation inevitably occur in the aeronautical components, especially the hot-section components [48]. Hot corrosion and high-temperature oxidation can degrade the material’s performance and reduce the service life of these components [49]. So the hot corrosion resistance and high-temperature oxidation resistance of superalloy must be enhanced.

To enhance the hot-corrosion resistance of the aeronautical components’ material, Cao et al. [50], selected Ni-based superalloy GH202 as experimental material and investigated the hot corrosion behavior of GH202 treated by LSP. In this work, the LSP experiment was performed in a Q-switched Nd3+:YAG laser with a repetition rate of 0.5 Hz and wavelength of 1.6 μm, and the hot corrosion environment was selected as 800 °C and 900 °C at molten salt. The corrosion kinetics curve of GH202 experimental before and after LSP treatment is displayed in Figure 7 [50]. Since oxidation films were formed on the material’s surface at high temperatures, the mass of experimental samples before and after LSP were increased. After the hot corrosion time was over 1h, all experimental samples’ mass were decreased. As observed from Figure 7, when the hot corrosion time is around 1 h, the mass gain of the experimental sample after the LSP treatment is less than that of the experiment sample before the LSP treatment. So it can be indicated that the mass loss of the experimental sample after the LSP treatment is less than that of the observed sample before the LSP treatment. So LSP treatment can significantly improve the hot corrosion resistance of GH202 superalloy.

Apart from superalloy GH202, the hot corrosion resistance of the Ni-based single-crystal superalloy was also increased. The hot corrosion behavior of Ni-based single-crystal superalloy treated by LSP was investigated by Geng et al. [46], and its hot corrosion kinetics is shown in Figure 8. In this experiment, the laser parameters were 7 J for laser pulse energy, 1064 nm for wavelength, 20 ns for pulse width, 3mm for spot diameter, and 50% for overlap rate. According to the given laser parameters and the Equation (4), the laser power density in this experiment is 4.95 GW/cm^2^. Figure 8 shows that the mass of the non-LSP and LSP experimental samples are increased, reflecting the negative effect of harsh conditions on the material. Interestingly, after two cycles, the mass gains of the LSP-treated experimental sample were more stable during the subsequent cycles. In contrast, the mass gains of non-LSP-treated experimental samples continue to increase.

Compared with Figure 7 and Figure 8, it can be seen that the phenomenon of superalloy GH202 represented weight loss, while the phenomenon of single-crystal superalloy represented weight addition. But the two phenomenons all reflect the improvement of the experimental sample’s hot corrosion resistance by the standard LSP treatment.

It is well known that the mechanical properties of near-surface layer can affect the corrosion behaviour of metallic materials components strongly [51]. The hot corrosion resistance improvement of the aeronautical components treated by LSP mainly result from the combine influences of the beneficial compressive residual stress, working hardening, grain refinement, and crystal defects (eg., high-density dislocations) in the near-surface layer [52,53,54]. The cross-section microstructures of superalloy GH202 experimental specimens before and after the LSP treatment characterized by EBSD is presented in Figure 9 [50]; the different colors represents different orientations. From the Figure 9a, it can be seen that the grains of the superalloy GH202 experimental sample before LSP treatment were distributed unevenly, and some annealing twins can be seen trough the grains. Since the high-pressure plasma shock waves induced by LSP treatment, an arc distribution of the grains which was close to the near-surface layer of superalloy GH202 experimental sample can be observed. Compared with the superalloy GH202 experimental sample before LSP treatment, the grains of superalloy GH202 experimental sample after the LSP treatment were refined significantly. In addition, the twins density for experimental sample after the LSP treatment was also increased significantly.

It is well known that the grain refinement of microstructures can lead to the increase of dislocation density, which will contribute to the improvement of microhardness. According to the Hall-Petch theory, the microhardness caused by grain refinement can be expressed as follows [55].
(10)$$H_{V}=H_{0}+\alpha Gb\rho$$
where *H*_V_ is the microhardness, *H*_0_ is the initial microhardness, *G* is the shear modulus, α is the material’s constants, *b* is the Burgers vector, and *ρ* is the dislocation density. Hence, LSP induce the dislocation density enhancement will contribute to enhance the microhardness in near-surface layer. In addition, the dislocation density enhancement will facilitate the formation of the homogeneous oxidation film and lead to the microstructure evolution, thereby providing the diffusion paths for the elements such as Cr, Al, and Ti, further producing a protective oxidation film. As a result, the hot corrosion resistance of metallic materials can be improved evidently after the LSP treatment [49,50,52].

In addition, some studies [54,55] have proved that the dislocations in near-surface layer plays an important role in the effect of grain refinement. The movement and accumulation for the dislocations can lead to the grain refinement in the near-surface layer of LSP treated samples, which also help to enhance the corrosion resistance. To reveal the fine structures of microstructures for the superalloy GH202 after the LSP treatment, the representative top surface of experimental sample after the LSP treatment was characterized by transmission electron microscopy (TEM) observation. The TEM image of top surface for superalloy GH202 experimental sample before and after the LSP treatment are shown in Figure 10 and Figure 11, respectively.

Compared with the superalloy GH202 experimental sample before the LSP treatment (Figure 10a), the dislocation density of the superalloy GH202 experimental samples after the LSP treatment was increased significantly (Figure 11a). In addition, the dislocation array can be seen in Figure 11a, which is caused by the reaction and motion of dislocations activated by the laser-induced plasma shock waves. The γ’ phases were dispersed evenly after the LSP treatment. At the same time, many twins were present on the surface of superalloy GH202 experimental sample after the LSP treatment, and a few dislocation lines through the twins were observed. And there are several dislocation tangles can be seen in the grain boundaries, which will lead to the limitation of the dislocation motion and the corrosion resistance improvement for the metallic materials or components [56,57].

## 4. Conclusions and Outlook

LSP is a novel surface-strengthening technology to modify microstructures and induce compressive residual stress on the near-surface layer of materials, thereby enhancing mechanical performance. The main LSP parameters mainly consists of laser parameters (pulsed laser energy, wavelength, laser pulse duration, laser beam spot diameter, and overlap rate), absorbing protective layer, and constrain layer. Select the suitable LSP parameters can obtain a better LSP effect. As an advanced mechanical surface-strengthening technology, LSP treatment shows tremendous advantages in improving mechanical performance for aeronautical components. This work summary the fundamental mechanism of LSP in detail and introduce several typical cases of applying LSP treatment to enhance the wear and corrosion resistance of aeronautical components materials, which will provide significant reference value and guiding significance for researchers to explore further the fundamental mechanism of LSP and the aspects of the wear and corrosion resistance extension of the aeronautical components.
(1)LSP utilizes the stress effect of laser-induced plasma shock waves to form a gradient distribution of compressive residual stress, microhardness, and microstructure in the near-surface layer of the metallic target, thereby improving the mechanical properties.(2)The wear resistance improvement of metallic material treated with the LSP treatment is attributed to the enhancement of microhardness and beneficial compressive residual stress.(3)The increased hot corrosion resistance of metallic materials treated by LSP is mainly attributed to the introduction of beneficial compressive residual stress, grain refinement, and crystal defects.

## Figures and Tables

**Figure 1 materials-16-04124-f001:**
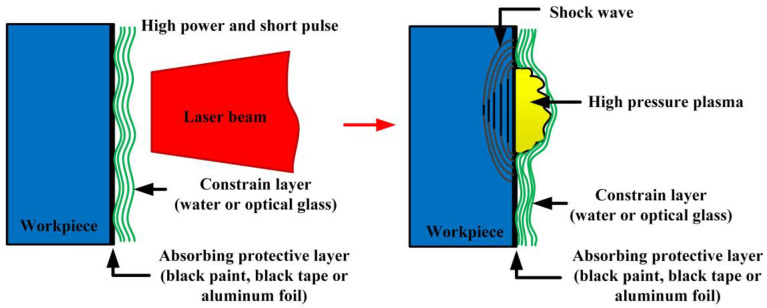
Fundamental mechanism schematic of LSP [29].

**Figure 2 materials-16-04124-f002:**
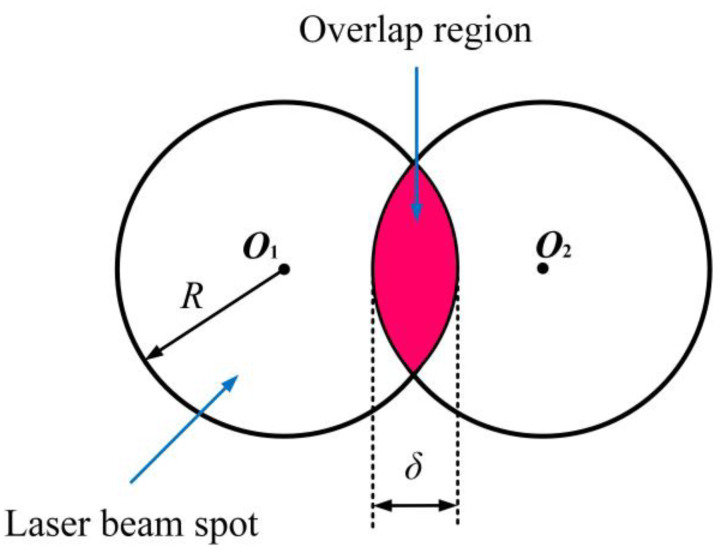
Schematic of overlap rate.

**Figure 3 materials-16-04124-f003:**
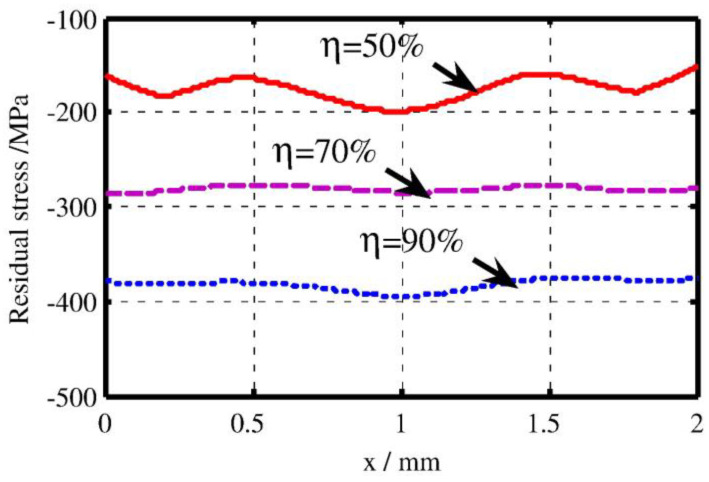
Residual stress distribution on the top surface for different overlap rates [39].

**Figure 4 materials-16-04124-f004:**
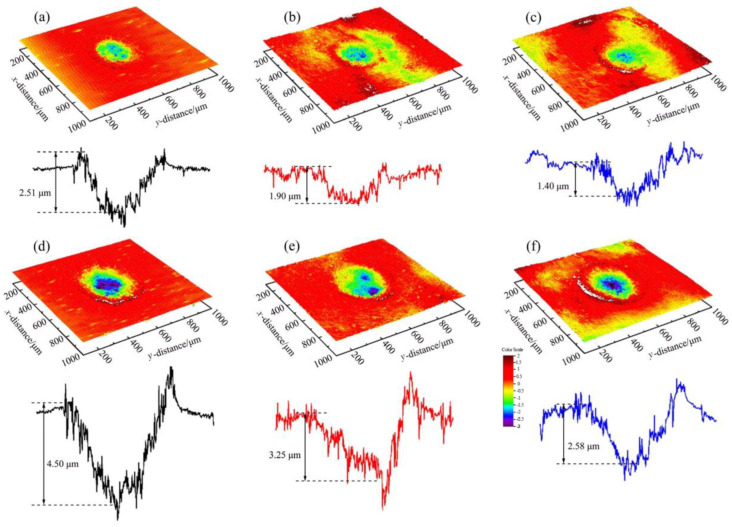
3D-profile micrographs of the worn scars of experimental samples in different impact wear test conditions: (**a**) prior LSP treatment with initial impact velocities of 60 mm/s; (**b**) 5 J LSP treatment with initial impact velocities of 60 mm/s; (**c**) 7 J LSP treatment with initial impact velocities of 60 mm/s; (**d**) prior LSP treatment with initial impact velocities of 150 mm/s; (**e**) 5 J LSP treatment with initial impact velocities of 150 mm/s; (**f**) 7 J LSP treatment with initial impact velocities of 150 mm/s [41].

**Figure 5 materials-16-04124-f005:**
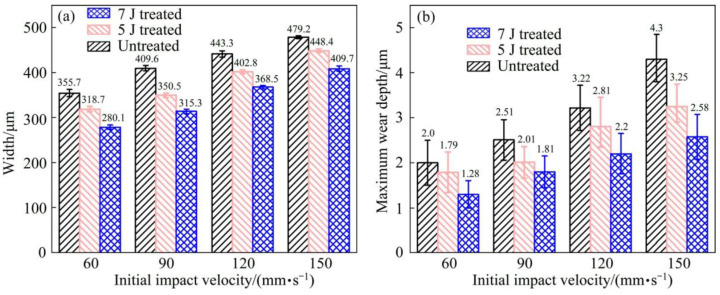
Width and maximum wear depth of wear scars for the experimental samples at different initial impact velocities: (**a**) width; (**b**) maximum wear depth [41].

**Figure 6 materials-16-04124-f006:**
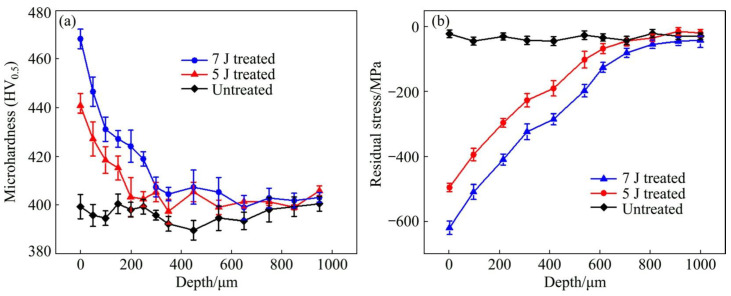
Microhardness and residual stress distribution of experimental samples before and after LSP treatment: (**a**) microhardness; (**b**) Residual stress [41].

**Figure 7 materials-16-04124-f007:**
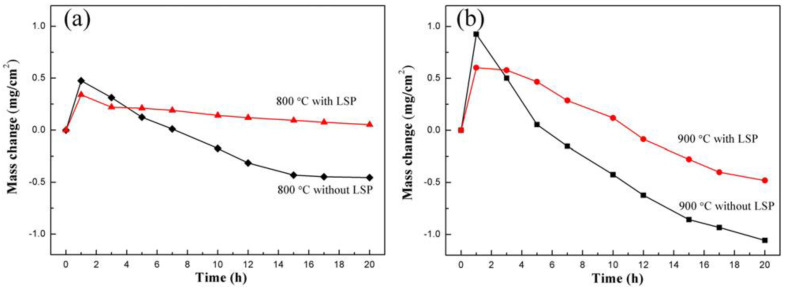
The corrosion kinetics curve of GH202 experimental before and after the LSP treatment: (**a**) at 800 °C; (**b**) at 900 °C [50].

**Figure 8 materials-16-04124-f008:**
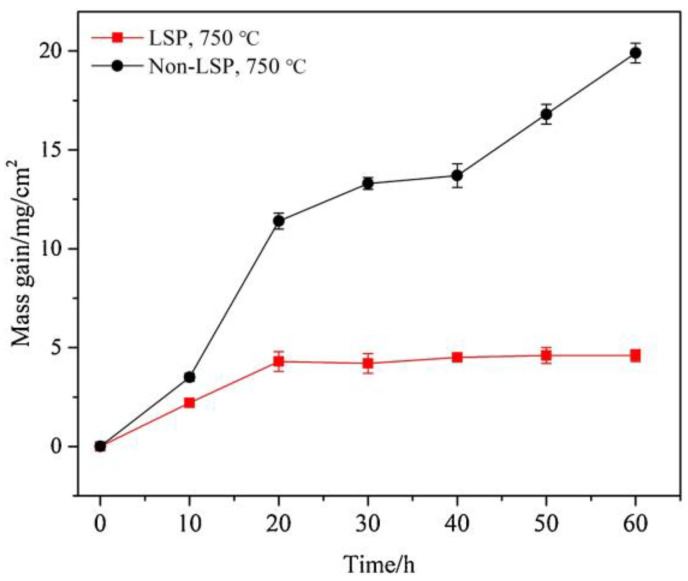
Mass changes of the Ni-based single-crystal superalloy experimental sample before and after the LSP treatment [46].

**Figure 9 materials-16-04124-f009:**
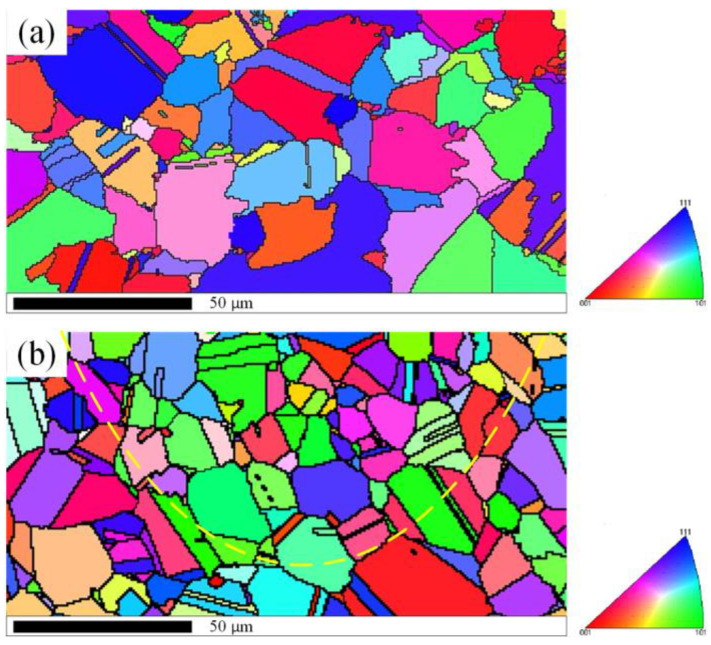
The cross-section microstructures of superalloy GH202 experimental specimens characterized by EBSD before and after the LSP treatment: (**a**) before the LSP treatment; (**b**) after the LSP treatment [50].

**Figure 10 materials-16-04124-f010:**
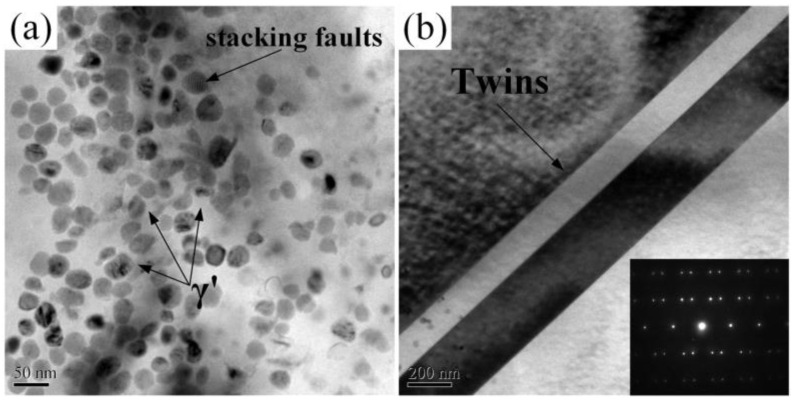
TEM image of top surface for superalloy GH202 experimental sample before the LSP treatment: (**a**) γ’ phase and stacking faults; (**b**) annealing twins [50].

**Figure 11 materials-16-04124-f011:**
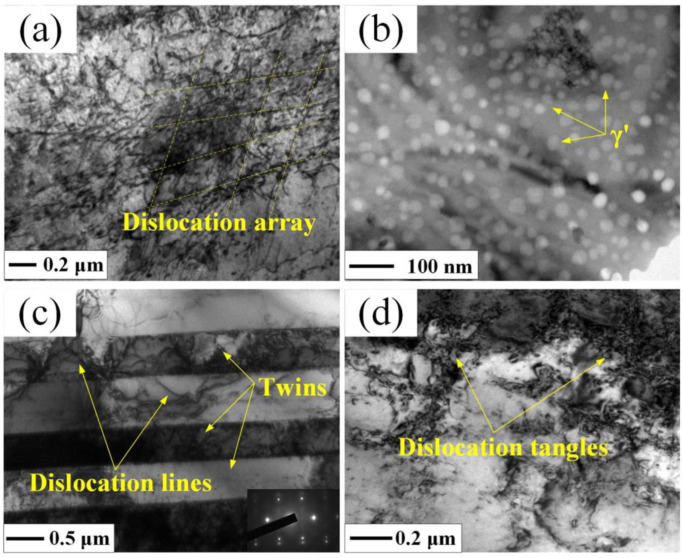
TEM image of top surface for superalloy GH202 experimental sample after the LSP treatment: (**a**) Dislocation array; (**b**) γ’ phases; (**c**) Twins and dislocation lines; (**d**) Dislocation tangles in the grain boundaries [50].

## Data Availability

All data supporting the conclusions of this manuscript are included within the manuscript.

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
