# Peer review of "Improving the Wear and Corrosion Resistance of Aeronautical Component Material by Laser Shock Processing: A Review"

_materials, 2023, doi:10.3390/ma16114124_

Round 1

Reviewer 1 Report

Dear authors,

I read your interesting review on "Improving the Wear and Corrosion Resistance of Aeronautical Component Material by Laser Shock Processing: A Review." Please reconsider the following items.

(m) mandatory

(o) optional

[1] (m) English: I have added it directly in the manuscript.

[2] (m) References: Please double-check the Instructions for Authors for the format of references.

https://www.mdpi.com/journal/materials/instructions

Best regard,

Author Response

# Reviewer: 1

I read your interesting review on "Improving the Wear and Corrosion Resistance of Aeronautical Component Material by Laser Shock Processing: A Review." Please reconsider the following items.

Question 1 (Q1): English: I have added it directly in the manuscript.

Answer 1 (A1): Thanks for your suggestions for improvement of our manuscript. According to your advice, we have checked our manuscript carefully, and all the revised parts have been marked in red.

Q2: References: Please double-check the Instructions for Authors for the format of references. https://www.mdpi.com/journal/materials/instructions.

A2: Thanks for your suggestions for improvement of our manuscript. All the format of the references in our revised manuscript have checked to meet the publication requirements.

Reviewer 2 Report

The Review deals with the Improving the Wear and Corrosion Resistance of Aeronautical

Component Material by Laser Shock Processing. 

According to the reviewer, the paper is worth publishing at Materials Journal, 

but some corrections are needed and then the paper can be accepted for publication in the journal.

While the authors have made considerable research effort, 

the presentation of the paper and the results must be proved. 

Additionally make the following corrections to the manuscript:

Comment 1

Line 24

aeronautical aeronautical component materials.

Two aeronautical aeronautical

Extended text editing.

Line 120

the pres-sure of

Extended text editing.

Line 142

is the static yield strengthen,

Extended text editing.

Figure 1

layer(water

The authors should replace

layer (water

Line 262

of these components[44]. So

The authors should replace (insert a space)

of these components [44]. So

Line 281

by Geng et al.[41], and

The authors should replace (insert a space)

by Geng et al. [41], and

Comment 2

Line 96

It's not so good to start the sub-section at the bottom of the page without using text.

Ιt is suggested that the authors format the text.

Comment 3

Lines 114 and 289

It is no so good to use the word "we".

The authors should rephrase.

Comment 4

Line 138

According to the theory of Johnson and Rhode,

The authors should insert a corresponding bibliographical reference.

Comment 5

The authors should provide Figure permission from the owner of the copyright of the original figure (usually the publisher) (for Figures: 1 - 6)

Comment 6

According to the journal's instructions:

Figures should be placed in the main text near to the first time they are cited.

The authors should format the text (remove the Figures).

Comment 7

References

The authors should delete the [J], [M] and [D].

The authors must format the References according to the journal's instructions

References should be described as follows, depending on the type of work:

Journal Articles:

1. Author 1, A.B.; Author 2, C.D. Title of the article. Abbreviated Journal Name Year, Volume, page range.

Comment 8

The authors should consider if they add a Figure with a real Laser Shock Processing device and

a Figure with real workpieces before and after Laser Shock Processing. 

Author Response

# Reviewer: 2 

The Review deals with the Improving the Wear and Corrosion Resistance of Aeronautical Component Material by Laser Shock Processing. According to the reviewer, the paper is worth publishing at Materials Journal, but some corrections are needed and then the paper can be accepted for publication in the journal. While the authors have made considerable research effort, the presentation of the paper and the results must be proved. Additionally make the following corrections to the manuscript:

Comment 1  

Line 24 aeronautical aeronautical component materials. Two aeronautical aeronautical Extended text editing.

Line 120  the pres-sure of Extended text editing.

 Line 142 is the static yield strengthen, Extended text editing.

Figure 1  layer(water

The authors should replace layer (water

Line 262

of these components[44]. So The authors should replace (insert a space) of these components [44]. So

Line 281 by Geng et al.[41], and The authors should replace (insert a space) by Geng et al. [41], and

A1: Thanks for your suggestions for improvement of our manuscript. According to your advice, we have revised our manuscript carefully. In addition, the Figure 1 has been replaced in this manuscript.

Comment 2

Line 96 It's not so good to start the sub-section at the bottom of the page without using text. Ιt is suggested that the authors format the text.

A2: Thanks for your suggestions for improvement of our manuscript. We have formatted the text, now the manuscript has become aesthetically pleasing.

Comment 3

Lines 114 and 289 It is no so good to use the word "we". The authors should rephrase.

A3: Thanks for your suggestions for improvement of our manuscript. We have rephrased.

Comment 4

Line 138 According to the theory of Johnson and Rhode, The authors should insert a corresponding bibliographical reference.

A4: Thanks for your suggestions for improvement of our manuscript. The corresponding bibliographical reference by Johnson and Rhode has been inserted in this revised manuscript.

Comment 5

The authors should provide Figure permission from the owner of the copyright of the original figure (usually the publisher) (for Figures: 1 - 6)

A5: Thanks for your suggestions for improvement of our manuscript. All the referred Figures has got permission, and the copyright has been upload in manuscript submission system.

Comment 6

According to the journal's instructions: Figures should be placed in the main text near to the first time they are cited. The authors should format the text (remove the Figures).

A6: Thanks for your suggestions for improvement of our manuscript. We have checked this manuscript to meet the requirement of the publication.

Comment 7

References 

The authors should delete the [J], [M] and [D].

The authors must format the References according to the journal's instructions

References should be described as follows, depending on the type of work:

Journal Articles:

  1. Author 1, A.B.; Author 2, C.D. Title of the article. Abbreviated Journal Name Year, Volume, page range.

A7: Thanks for your suggestions for improvement of our manuscript. All the format of the references in our revised manuscript have checked to meet the publication requirements.

Comment 8

The authors should consider if they add a Figure with a real Laser Shock Processing device and a Figure with real workpieces before and after Laser Shock Processing.

A8: Thanks for your suggestions for improvement of our manuscript. Since the experiment was referred the precious publications, so we cannot provide the real LSP device. In addition, the real workpieces before and after LSP can not found in the listed references. So we are very sorry that we cannot add the advice Figure.

Reviewer 3 Report

This is the review paper concerning with the application of Laser Shock Processing (LSP) to improve the wear and corrosion resistance of materials used in aeronautical components. The authors are analyzing 47 papers from the year span 1970-2023, the focus being on newer papers.

This is a technically sound paper, and I have only a couple of comments: (1) How the authors have chosen the papers for their analysis? It seems that no detailed bibliographical analysis have been carried out. (2) How covering is this review? There are only a few alloys named in this paper. I would like to see, in the beginning of Section 3, a summarizing table to show which of the listed references deals with which property (wear or corrosion) (3) The authors say both in Introduction and Conclusions, that one aim of the paper is to provide guidelines for researchers. Are the authors able to define the future research directions in more details? After this review, they should.

There are much too many typos and grammatical errors in the paper, and a proper language check by the native speaker is needed.

Author Response

# Reviewer: 3

This is the review paper concerning with the application of Laser Shock Processing (LSP) to improve the wear and corrosion resistance of materials used in aeronautical components. The authors are analyzing 47 papers from the year span 1970-2023, the focus being on newer papers.

This is a technically sound paper, and I have only a couple of comments: (1) How the authors have chosen the papers for their analysis? It seems that no detailed bibliographical analysis have been carried out. (2) How covering is this review? There are only a few alloys named in this paper. I would like to see, in the beginning of Section 3, a summarizing table to show which of the listed references deals with which property (wear or corrosion) (3) The authors say both in Introduction and Conclusions, that one aim of the paper is to provide guidelines for researchers. Are the authors able to define the future research directions in more details? After this review, they should.

There are much too many typos and grammatical errors in the paper, and a proper language check by the native speaker is needed.

Answer: Thanks for your suggestions for improvement of our manuscript. According to your advice, we have checked our manuscript carefully, and all the revised parts have been marked in red.

Reviewer 4 Report

The article presents the descriptions of the technology: Laser Shock Processing (LSP). Authors written that is a novel surface strengthening technology to modify microstructures and induce beneficial compressive residual stress on the near-surface layer of materials, thereby enhancing mechanical performance. This paper will provide very important reference value and guiding significance for researchers to further explore the fundamental mechanism of LSP and the aspects of the wear and corrosion resistance extension of the aeronautical components.

However, the article needs to be corrected:

Lines 139 and 143;
  Equations are wrongly numbered.

Describe in the article the measurement methodology.

Specify how microhardness was measured (standard, number of measurements, hardness tester, etc.).

How were residual stresses measured?

What wear test was performed?

Author Response

# Reviewer: 4

The article presents the descriptions of the technology: Laser Shock Processing (LSP). Authors written that is a novel surface strengthening technology to modify microstructures and induce beneficial compressive residual stress on the near-surface layer of materials, thereby enhancing mechanical performance. This paper will provide very important reference value and guiding significance for researchers to further explore the fundamental mechanism of LSP and the aspects of the wear and corrosion resistance extension of the aeronautical components.

 However, the article needs to be corrected:

Lines 139 and 143; Equations are wrongly numbered.
A1: Thanks for your suggestions for improvement of our manuscript. The equations number has been checked.

Describe in the article the measurement methodology.
Specify how microhardness was measured (standard, number of measurements, hardness tester, etc.).
How were residual stresses measured?
What wear test was performed?

A2: Thanks for your suggestions for improvement of our manuscript. The measurement methodology has added in this manuscript. And the revised parts were marked in red in this revised manuscript.

Reviewer 5 Report

The article provides basic information about LSP technology and a brief overview of its parameters. The literature review is numerous.
I have some fundamental comments on the paper:
- As a review article on the benefits of LSP technology, it is relatively brief. The introduction often mentions the fatigue properties of materials used in aerospace, which are improved and service life extended by the application of LSP. However, the results of experimental studies are not provided by the authors and the main part of the review does not mention this aspect at all
- The potential of LSP to reduce wear is undisputed and well documented and physically supported in the review - this cannot be said of the increase in corrosion resistance - a deeper physical analysis of the processes is lacking
- The article would have benefited from a concise and illustrative overview and specifics of component stresses in aerospace and aircraft engines
- The literature would be expanded to include articles from the International Journal of Fatigue. Material Science and Engineering, and others, which will broaden the reader's horizon and the literature review thus supplemented will be more relevant to the topic described
- The influence of LSP on fatigue properties can certainly be found in the above mentioned journals and the review article supplemented
- In the conclusions, it would be useful to add and graphically interpret the areas of improvement in material properties due to the application of LSP and, if known, the degree of improvement, and also to summarize the basic parameters of the LSP technology at the beginning of the conclusions section

Author Response

The article provides basic information about LSP technology and a brief overview of its parameters. The literature review is numerous. I have some fundamental comments on the paper: Question 1 (Q1): As a review article on the benefits of LSP technology, it is relatively brief. The introduction often mentions the fatigue properties of materials used in aerospace, which are improved and service life extended by the application of LSP. However, the results of experimental studies are not provided by the authors and the main part of the review does not mention this aspect at all. Answer 1 (A1): Thanks for your suggestions for improvement of our manuscript. In fact, this manuscript is aim to reviewed the wear and corrosion resistance improvement of aeronautical components by LSP, so the results of experimental studies about fatigue properties are not provided in this manuscript. So we have clarified the main purpose of this paper in the section of introduction, and the related revised parts were marked in red. Q2:The potential of LSP to reduce wear is undisputed and well documented and physically supported in the review - this cannot be said of the increase in corrosion resistance - a deeper physical analysis of the processes is lacking. A2: Thanks for your suggestions for improvement of our manuscript. We added some contents about the deeper physical analysis about the corrosion resistance improvement, and the added/revised parts were marked in red in the revised manuscript. Q3:The article would have benefited from a concise and illustrative overview and specifics of component stresses in aerospace and aircraft engines. A3: Thanks for your suggestions for improvement of our manuscript. This revised manuscript added some content about the component stresses such as centrifugal stress, thermal stress, aerodynamic loading, vibration load. And the revised content was marked in red. Q4:The literature would be expanded to include articles from the International Journal of Fatigue. Material Science and Engineering, and others, which will broaden the reader's horizon and the literature review thus supplemented will be more relevant to the topic described. A4: Thanks for your suggestions for improvement of our manuscript. We have added some related references in this revised manuscript, and all the revised/added reference were marked in red. Q5:The influence of LSP on fatigue properties can certainly be found in the above mentioned journals and the review article supplemented. A5: Thanks for your suggestions for improvement of our manuscript. This work is aim to review the wear and corrosion resistance improvement by LSP. So the content about the influence of LSP on fatigue properties is less in relative. Q6: In the conclusions, it would be useful to add and graphically interpret the areas of improvement in material properties due to the application of LSP and, if known, the degree of improvement, and also to summarize the basic parameters of the LSP technology at the beginning of the conclusions section. A6: Thanks for your suggestions for improvement of our manuscript. We have added content to summary the basic parameter of the LSP at the section of conclusions.

Reviewer 6 Report

The authors provided a review on the effect of laser shot peening on wear and corrosion resistance properties. While the manuscript is generally well executed, there are several issues that should be addressed before further consideration for publication.

1. Suggest the authors to discuss the key parameters involved in laser shot peening and how they can affect the results.

2. Are the effects material specific? Or they can be applied to any materials? Any discussion on this?

3. What about the challenges and potential of this approach? How do they compare to other techniques?

Author Response

The authors provided a review on the effect of laser shot peening on wear and corrosion resistance properties. While the manuscript is generally well executed, there are several issues that should be addressed before further consideration for publication. Q1: Suggest the authors to discuss the key parameters involved in laser shot peening and how they can affect the results. A1: Thanks for your suggestions for improvement of our manuscript. We have added some contents to introduce the key parameters of LSP mainly in section 2, and the added contents were marked in red. Q2: Are the effects material specific? Or they can be applied to any materials? Any discussion on this? A2: Thanks for your suggestions for improvement of our manuscript. The LSP effect can be only induced in the metallic materials. As an advanced surface modify technology, LSP has been applied in many area. Since this manuscript is aim to review the wear and corrosion resistance improvement by LSP, so the other discussions is not summarized in this work. Q3: What about the challenges and potential of this approach? How do they compare to other techniques? A3: Thanks for your suggestions for improvement of our manuscript. Compared with other techniques SP, URSS, USP, and WJP, the adequate beneficial compressive residual stress layer depth induced by LSP treatment can reach over 1 mm. In addition, LSP treatment can obtain more smooth surface morphology relative; that is, LSP can obtain a lower macroscopic surface roughness. Despite these advantages, the efficiency of LSP is very low, and its cost is very high. In addition, the intelligence degree of LSP equipment is low in relative.

Round 2

Reviewer 1 Report

Good!

Author Response

Thanks for your suggestions for improvement of our manuscript.

Reviewer 2 Report

Comment 1

Line 96

It's not so good to start the sub-section at the bottom of the page without using text.

Ιt is suggested that the authors format the text.

Comment 2

According to the journal's instructions:

Figures should be placed in the main text near to the first time they are cited.

The authors should format the text (remove the Figures 1, 2 and 3).

Comment 3

Line 203

residual stress [39-39]. And the 

The authors must check ([38-39]?)

Line 267

Cao et al. [45] selected

The authors must check (Cao et al. [46]?)

Author Response

Comment 1

Line 96

It's not so good to start the sub-section at the bottom of the page without using text.

Ιt is suggested that the authors format the text.

A1: Thanks for your suggestions for improvement of our manuscript. We have formatted the text, now the problem about the sub-section at the bottom of the page has been solved.

Comment 2

According to the journal's instructions:

Figures should be placed in the main text near to the first time they are cited.

The authors should format the text (remove the Figures 1, 2 and 3).

A2: Thanks for your suggestions for improvement of our manuscript. We have formatted the text (Figures 1, 2, and 3).

Comment 3

Line 203

residual stress [39-39]. And the 

The authors must check ([38-39]?)

 Line 267

Cao et al. [45] selected

The authors must check (Cao et al. [46]?)

A3: Thanks for your suggestions for improvement of our manuscript. The problems about reference number has been solved.

Reviewer 3 Report

This is the revised version of the paper that I have reviewed earlier. It seems that the authors are either unable or unwilling to answer to my quite simple questions. Knowing, how this kind of review has been carried out increases the reader's confidence, and explains the distribution of paper that has been referred to. In this case, there are quite a small number of refernces on widely studied research field.

I do not want to delay the publication of the technically sound paper any more, and recommend publishinf it as it is,.

Author Response

Answer: Thanks for your suggestions for improvement of our manuscript.

Reviewer 5 Report

The authors significantly increased the quality of the paper - the reviewers' work was meaningless. On line 196 "microhArdness" watch out for spelling...

Author Response

Question 1 (Q1):The authors significantly increased the quality of the paper - the reviewers' work was meaningless. On line 196 "microhArdness" watch out for spelling...

Answer 1 (A1): Thanks for your suggestions for improvement of our manuscript. We have the spelling of “microhardness” on line 196, and the revised content was marked in red.

Other revision

A: All other revision in this revised manuscript were marked in red.

Reviewer 6 Report

NIL

Author Response

Question 1 (Q1): NIL

Answer 1 (A1): Thanks for your suggestions for improvement of our manuscript. We will continue to revise this manuscript to fit the publication requirement of Materials.

Other revision

A: All other revision in this revised manuscript were marked in red.